# Improving the Spatiotemporal Transferability of Hyperspectral Remote Sensing for Estimating Soil Organic Matter by Minimizing the Coupling Effect of Soil Physical Properties on the Spectrum: A Case Study in Northeast China

**Yuanyuan Sui** [1], **Ranzhe Jiang** [1], **Nan Lin** [2,*], **Haiye Yu** [1] and **Xin Zhang** [1]

1   College of Biological and Agricultural Engineering, Jilin University, Changchun 130012, China;
    suiyuan@jlu.edu.cn (Y.S.); jiangrz23@mails.jlu.edu.cn (R.J.); haiye@jlu.edu.cn (H.Y.); zhangx@jlu.edu.cn (X.Z.)
2   School of Geomatics and Prospecting Engineering, Jilin Jianzhu University, Changchun 130118, China
*   Correspondence: linnan@jlju.edu.cn

**Abstract:** Soil organic matter (SOM) is important for the global carbon cycle, and hyperspectral remote sensing has proven to be a promising method for fast SOM content estimation. However, because of the neglect of the spectral response of soil physical properties, the accuracy and spatiotemporal transferability of the SOM prediction model are poor. This study aims to improve the spatiotemporal transferability of the SOM prediction model by alleviating the coupling effect of soil physical properties on spectra. Based on satellite hyperspectral images and soil physical variables, including soil moisture (SM), soil surface roughness (root-mean-square height, RMSH), and soil bulk weight (SBW), a soil spectral correction model was established based on the information unmixing method. Two important grain-producing areas in Northeast China were selected as study areas to verify the performance and transferability of the spectral correction model and SOM content prediction model. The results showed that soil spectral corrections based on fourth-order polynomials and the XG-Boost algorithm had excellent accuracy and generalization ability, with residual predictive deviations (RPDs) exceeding 1.4 in almost all the bands. In addition, when the soil spectral correction strategy was adopted, the accuracy of the SOM prediction model and the generalization ability after the model migration were significantly improved. The SOM prediction accuracy based on the XG-Boost-corrected spectrum was the highest, with a coefficient of determination ($R^2$) of 0.76, a root-mean-square error (RMSE) of 5.74 g/kg, and an RPD of 1.68. The prediction accuracy, $R^2$ value, RMSE, and RPD of the model after the migration were 0.72, 6.71 g/kg, and 1.53, respectively. Compared with the direct migration prediction of the model, adopting the soil spectral correction model based on fourth-order polynomials and XG-Boost reduced the RMSE of the SOM prediction results by 57.90% and 60.27%, respectively. This performance comparison highlighted the advantages for considering soil physical properties in regional-scale SOM predictions.

**Keywords:** soil organic matter; soil physical properties; hyperspectral imagery; spectrum correction; model migration

## 1. Introduction

With social development and the continuous growth of population and consumption, global food security issues have become increasingly prominent [1]. Accurately monitoring soil conditions is of great significance for ensuring sustainable agricultural development and global food security [2–4]. Soil organic matter (SOM) is an important component of soil, which not only provides nutrients for plant growth but also improves soil physical, chemical, and biological conditions [5]. Obtaining the spatial distribution information of the SOM in cultivated land is a key prerequisite for the utilization and management of soil resources and can provide key basic data for agricultural disaster assessment and food

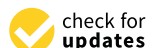



security scenario simulations [6,7]. In addition, a large portion of the SOM is composed of organic carbon, which constitutes the largest carbon pool in terrestrial ecosystems [8]. Minor soil carbon pool changes can significantly alter the atmospheric $CO_2$ concentration, thereby affecting global carbon cycling and the climate [9]. Therefore, precisely understanding the SOM content and spatial distribution is crucial for promoting sustainable agricultural development, enhancing the soil carbon sequestration potential, and regulating global climate change [10,11].

Remote sensing is a low-cost, high-accuracy, real-time method for multi-angle, multi-temporal, and large-area Earth observations [12–14]. To date, the ability of hyperspectral remote sensing to predict and map SOM contents has been confirmed in many studies [15–17]. Imaging spectra are the most important data source, and their characteristic response to the soil chemical composition is an important foundation for hyperspectral remote-sensing-based SOM content prediction [18–20]. As an important dyeing material of the soil, SOM can significantly absorb the incident light in the visible-light band, or even the short-wave infrared band, and there is usually a significant negative correlation between its content and the soil spectral reflectance. However, imaging spectra are not affected by the soil composition alone but comprehensively reflect the soil physical properties and chemical composition within the ground sample, and the soil physical properties and chemical composition exert a coupling effect on the response to the spectrum [21–23]. Previous studies on SOM prediction lack consideration of soil physical properties, ignoring the spectral response of soil physical properties, such as soil moisture (SM), soil bulk weight (SBW), and surface roughness properties (e.g., root-mean-square height, RMSH), resulting in poor accuracy and poor spatiotemporal transferability of SOM prediction models. Research has shown that the scattering contribution of soil physical properties to spectral reflectance severely affects the sensitivity of hyperspectral data to the SOM content [24]. As the SM content increases continuously until saturation, the soil reflectance decreases first and then increases because of the specular reflection effect [25]. In addition, RMSH increments enhance light scattering and transmission on the soil surface, thus decreasing the reflectivity, especially in the visible and near-infrared wavelength ranges [26]. Meanwhile, long-term high-intensity mechanized planting increases the SBW of cultivated land. The concomitant changes in SM, SBW, and spectral reflectance exhibit a complex relationship. Generally, increases in SM, SBW, and RMSH decrease the spectral reflectance, showing a coupling effect [27,28]. Noteworthily, the effect of the SOM content on the soil spectrum is far weaker than that of soil physical properties [29]. Because of the satellite revisit period and soil physical condition uncertainties, the effect of the noise interference of soil physical properties on the spectrum greatly limits the accuracy and spatiotemporal transferability of the remote-sensing-based SOM evaluation model, which needs to be solved urgently.

Reducing the influence of soil physical properties on hyperspectral data is the key to improving the spatiotemporal transferability of SOM content prediction models. To this end, some scholars have tried to develop satellite hyperspectral image correction methods considering soil physical properties at the pixel scale to reduce the sensitivity of SOM prediction models to spectral differences caused by soil physical property changes [30]. Minasny et al. developed a spectral correction model based on the EOP method to eliminate the influence of the SM [31]. Ou et al. established a spectral correction model for SM removal based on the Kubelka–Munk method to improve the accuracy of SOM predictions [27]. Castaldi et al. synthesized the dry-soil spectrum by calculating the statistical variabilities of dry and wet soils, thereby improving the model prediction accuracy [21]. Although these methods have corrected hyperspectral data to a certain extent, they are mostly based on one physical parameter and ignore the coupling response of different soil physical properties to the spectrum. In particular, in cultivated soil affected by both natural and human factors, the complex physical properties of the soil have significant spatial and temporal heterogeneities and transient variability [20,32,33]. Spectral correction models based on a single soil physical parameter have limited improvements in SOM prediction accuracy and spatiotemporal transferability. To date, because of the scarcity of soil physical

data from satellite ground synchronization experiments, the potential of hyperspectral correction methods comprehensively considering multiple soil physical properties has not been fully explored.

Developing a soil spectral correction method by alleviating the coupling effect of surface physical properties on soil pixel spectra is an effective solution to improve the spatiotemporal transferability of the SOM prediction model. Complex interactions between various surface physical properties and electromagnetic waves make it difficult to simulate the relationship between soil physical properties and the spectrum with physical models [27]. Therefore, this study aims to separate the physical and chemical soil information in the spectral data with data-driven methods. Some of the issues that must be addressed in advance include (i) quantifying the effect of soil physical properties on satellite hyperspectral data and (ii) generalizing the model in areas with scarce soil sample data. Studies have indicated various functional relationships between soil physical parameters (SM, RMSH, and SBW) and spectral reflectance, including logarithmic, exponential, and power exponential functions [26,34]. However, most previous studies were based on multispectral data, while the effect of soil physical properties on reflectance remain to be clarified based on hyperspectral data with more continuous and dense bands [35,36]. Although these functional relationships may change slightly because of differences in soil types, components, etc., they generally reflect the spectral characteristics induced by soil physical properties and are an essential basis for representing soil physical property information in spectral data [37–39]. Based on this, spectral forward modeling can be carried out using soil physical properties, providing prior data for soil spectral correction [40]. Moreover, to effectively decompose the soil physical and chemical information in the pixel spectrum, nonlinear parameter regression and machine-learning models have been used to simulate the coupling relationship between soil physical property spectra and pixel spectra, respectively. These two data-driven methods search for the rules between data through statistical analysis and machine-learning training, respectively. In this way, unknown data can be predicted, and the generalization ability of the soil spectral correction method can be guaranteed by the regression equation. For the SOM prediction model, the soil spectral correction redistributes the observational information of the original hyperspectral remote sensing to ensure the uniformity of the soil physical properties of all the pixels, which helps to improve the spatiotemporal transferability of the model [41–43].

This work seeks to establish a hyperspectral SOM prediction model with high spatiotemporal transferability, which can guide soil investigation and parameter prediction. The objectives of this study are (i) evaluating the impact of soil physical properties on the image pixel spectrum and their contribution to the bias in SOM content prediction, (ii) developing soil spectral correction methods by alleviating the coupling impact of soil physical properties on the satellite spectrum, and (iii) determining the spatiotemporal transferability potential of satellite hyperspectral data for SOM retrieval based on a soil spectral correction strategy. Data-poor regions might benefit from the proposed SOM prediction model with strong spatiotemporal transferability when mapping the SOM to develop appropriate policies.

## 2. Materials and Methods

This study comprehensively considered the coupling effects of soil physical properties, such as SM, RMSH, and SBW, on satellite hyperspectral images and proposed a soil spectral correction method. The workflow includes data collection, spectral analysis, spectral correction, and modeling evaluation, as shown in Figure 1. First, parameter estimation equations were used to establish empirical relationships between satellite hyperspectral data and the three main soil physical parameters: SM, RMSH, and SBW. The spectral data of the soil physical properties were obtained by spectral forward simulation using soil physical parameters. Then, a soil pixel spectral correction model was constructed based on the simulated spectrum, soil pixel spectrum, and ground spectrum using multi-order polynomials and various machine-learning models to remove the effects of the SM, RMSH,

and SBW on the image spectrum. Finally, the SOM prediction model was constructed using XG-Boost, and the performance of the soil spectral correction model and its improvement effect on the SOM prediction accuracy were evaluated at two sites.

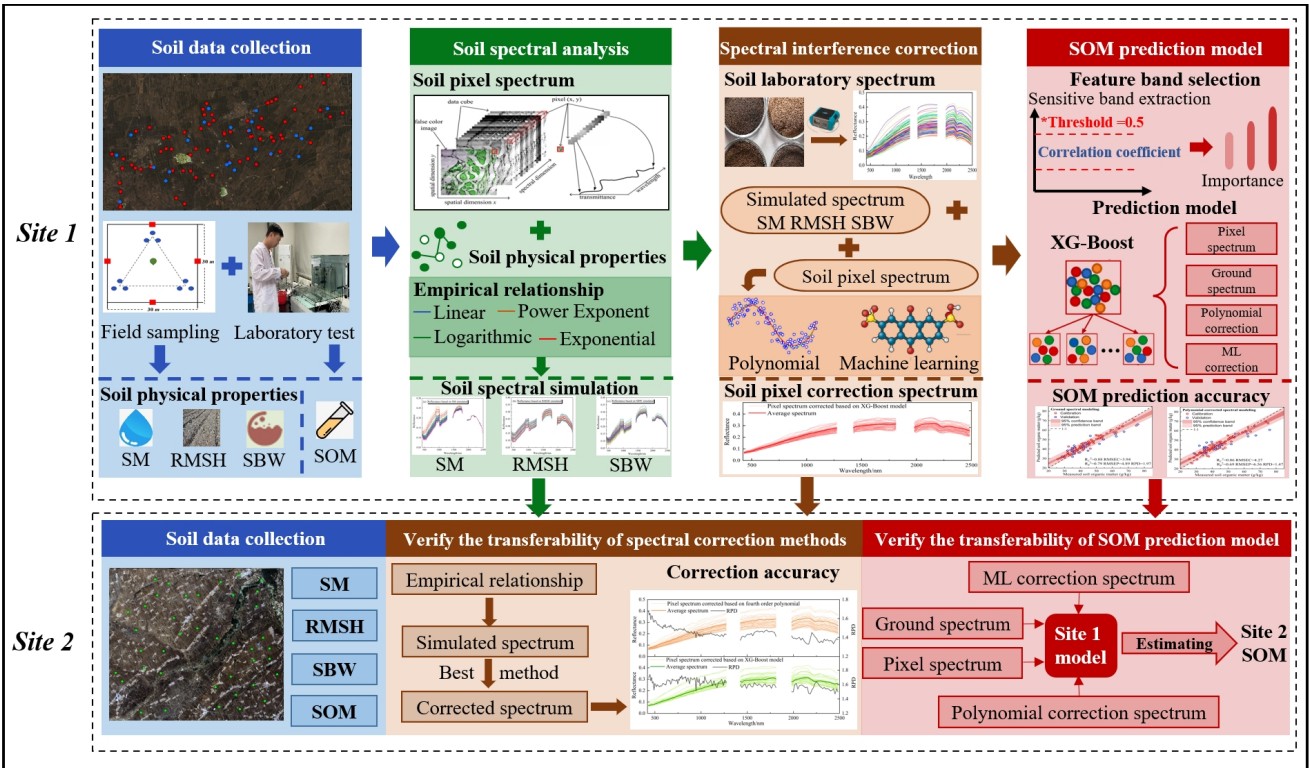

**Figure 1.** Framework of the proposed SOM estimation model.

### 2.1. Study Area

Site 1 is in the protected black soil cultivated land of Heilongjiang Province, Northeast China (131°30′–132°03′ E, 46°36′–46°49′ N), as shown in Figure 2, which has an area of 1095 km². This region features a temperate continental monsoon climate, with an annual precipitation of 450–650 mm. The precipitation is mainly concentrated in June–September, accounting for 80% of the annual total. The terrain of the study area is high in the south and low in the north, higher in the west than in the east, and most of the area is an accumulation plain [44]. This research area is in one of only four black soil areas in the world, characterized by a deep cultivated layer and fertile soil. The thickness of the soil layer containing humus is 25–80 cm, suitable for planting crops such as corn and soybeans [45].

Site 2 is in the protected black soil cultivated land of Jilin Province, China (125°24′–125°43′ E, 44°36′–44°46′ N), as shown in Figure 2, which has an area of 713 km². Its terrain is flat, with an elevation between 189 and 237 m. This region is a transitional zone between the humid mountainous area in the east and the semi-arid plain area in the west. This study area features a temperate continental semi-humid monsoon climate, with an average annual temperature of 4.6 °C and an annual precipitation of 600–700 mm. This region has rich river systems, relatively abundant agricultural water resources, and strong SM spatial heterogeneity. The soil in this region is mainly phaeozems with a 0.6–1.0 m humus-containing layer [46,47]. Site 2 has a significantly different soil type, surface characteristics, and other environmental factors than Site 1, which can verify the spatiotemporal transferability of the SOM content prediction model in this study.

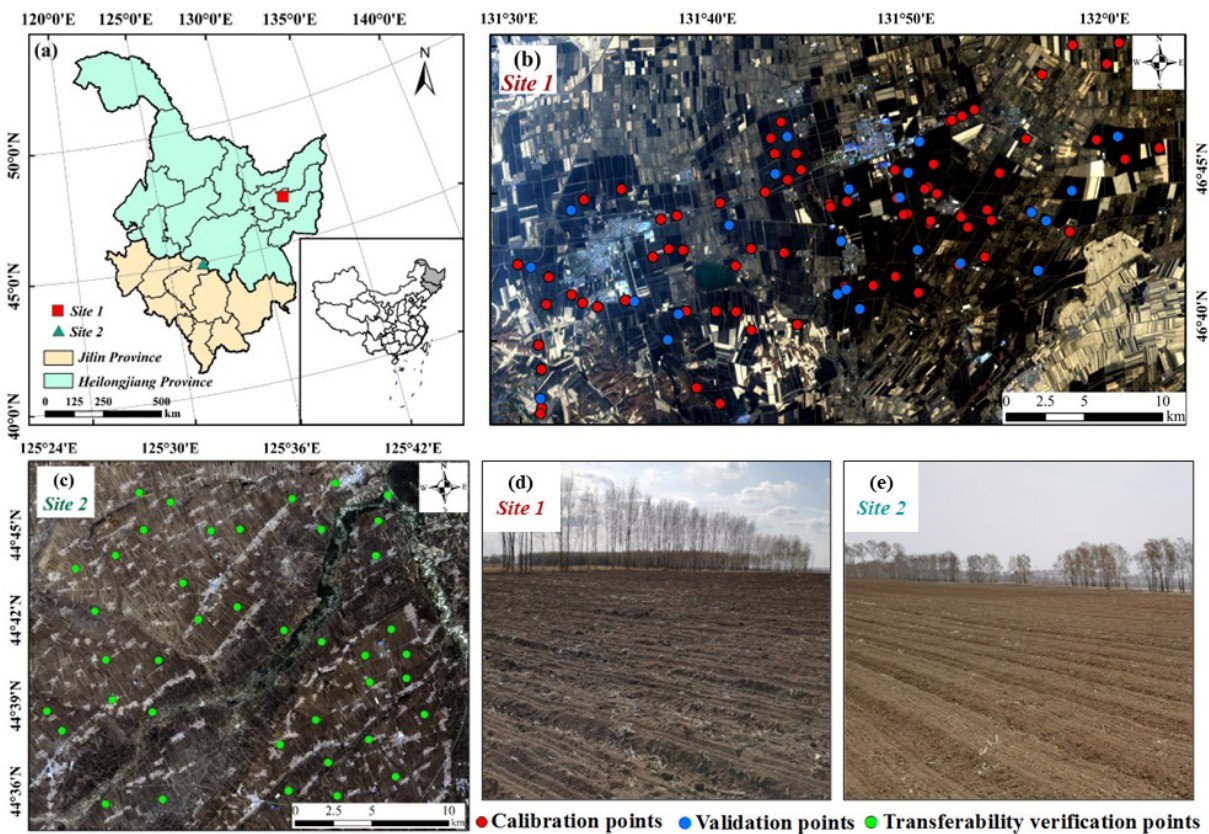

● **Calibration points**　　● **Validation points**　　● **Transferability verification points**

**Figure 2.** Study area overview. (**a**) The geographical location of the study area; (**b**,**c**) the soil-sampling points at Site 1 and Site 2, respectively; (**d**,**e**) the soil surfaces during the "bare soil period".

### 2.2. Datasets

### 2.2.1. Soil Sample Collection and Treatment

A total of 104 surface soil samples were collected from Site 1 between 29 and 30 October 2022 (Figure 2b). Among them, 80 soil samples were used as the training set for the models, and the remaining 24 samples were used as the validation set. Between 14 and 15 April 2023, 40 surface soil samples were collected from Site 2 (Figure 2c) for testing the spatiotemporal transferability of the models. First, a 3D laser scanner (Trimble TX8, Trimble, CA, USA; scanning speed: 1 million points per second) was installed at the midpoint of each edge in the quadrant to scan the soil surface structure (Figure 3). Next, nine subsamples were collected with a ring knife (depth: 5 cm and volume: 200 mL) in each 30 × 30 m quadrant. The real-time kinematic (RTK) survey technique was used to record the longitude and latitude of the quadrant's midpoint.

After the samples were transported to the laboratory, the SM and SBW of the nine subsamples from each quadrant were obtained through weighing and drying, and the average of the subsamples was calculated. Then, the nine subsamples were mixed into a composite sample for laboratory spectral measurements (ASD FieldSpec 4 spectrometer, ASD Specialty Healthcare, LLC, Sioux Falls, SD, USA) (taking the average of 10 measurements) and SOM content determination using the potassium dichromate heating method [48]. To ensure the same SBW of each sample, soil samples were loaded in a disposable culture dish for spectral measurements. The soil surface point cloud data of each measurement site were spliced, cropped, and filtered. The processed point cloud data were used to establish a three-dimensional relative coordinate system (Figure 3b); the Z coordinates of all the point cloud data were extracted, and the RMSH of the quadrant was calculated.

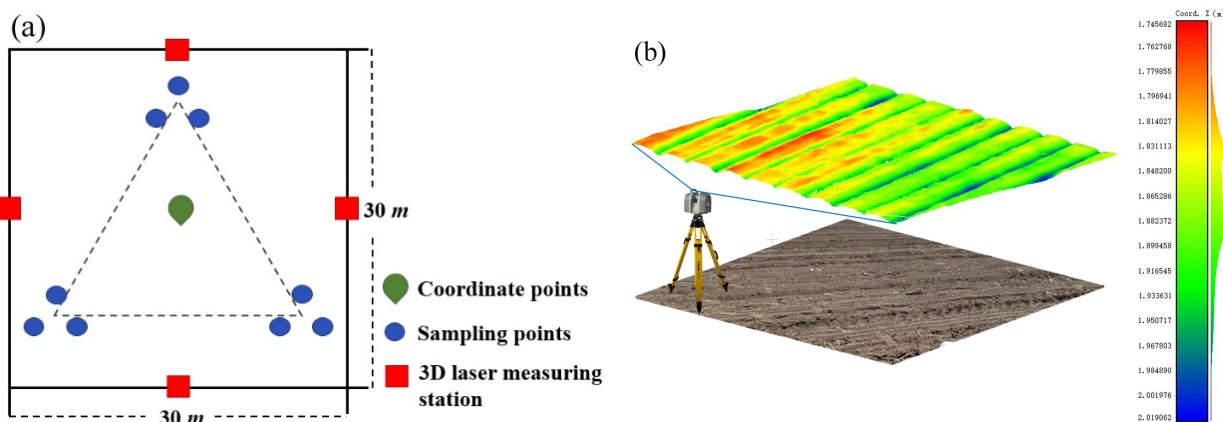

**Figure 3.** The schematic diagram of the soil sample collection and parameter measurement in the sample square. (**a**) Schematic diagram of the sampling strategy in the quadrant; (**b**) soil surface point cloud data measurement.

### 2.2.2. Image Acquisition and Treatment

The Ziyuan-1-02D (ZY1-02D) hyperspectral image data were acquired from the Aerospace Information Research Institute at the Chinese Academy of Sciences. The image generation time is synchronized with the soil-sampling time. All the images have less than 1% cloud coverage. The technical parameters of the ZY1-02D satellite hyperspectral camera are listed in Table 1. The sensor exhibits strong noises in wavelength ranges from 400 to 450 nm and from 2460 to 2500 nm and is affected by atmospheric water vapor absorption in wavelength ranges from 1290 to 1408 nm and from 1828 to 1963 nm [49]. Therefore, the 450 to 1290 nm, 1408 to 1828 nm, and 1963 to 2460 nm bands were selected as the spectral bands in this study. The images were subjected to stripe removal, geometric correction, and atmospheric correction. To correct the bidirectional reflectance distribution function effect of the image, the zenith angle and azimuth angle of the sun (and satellite) are calculated, and the image is corrected by the kernel-driven bidirectional reflectance distribution function model [50].

**Table 1.** ZY1-02D satellite hyperspectral camera parameters.

| Specification | Parameter |
| :---: | :---: |
| Spectral range (nm) | 400–2500 |
| Channels | 76 (VNIR); 90 (SWIR) |
| Spectral resolution (nm) | 10 (VNIR); 20 (SWIR) |
| Swath width (km) | 60 |
| Spatial resolution (m) | 30 |
| Revisit cycle (d) | 3 |
| Lateral swing capacity (°) | ±26 |

### 2.3. Spectral Correction Strategy

The image pixel spectrum comprehensively reflects the soil physical properties (e.g., SM, RMSH, and SBW) and chemical composition within the ground quadrant. Spectral correction aims to separate reflection features attributed to physical and chemical properties of the soil in pixel spectral data, thus alleviating the coupling effect of the soil physical properties on the spectrum. First, linear, exponential, power exponential, and logarithmic parameter estimation equations were used to establish empirical relationships between satellite hyperspectral data and the three soil physical parameters, SM, RMSH, and SBW, on a band-by-band basis. These parameter estimation methods for fitting the relationship between the soil physical properties and spectral reflectance have been verified in several studies [34,36].

Parameter estimation equations were used to establish empirical relationships between satellite hyperspectral data and the three main soil physical parameters: SM, RMSH, and

SBW. The spectral data of the soil physical properties were obtained by spectral forward simulation using soil physical parameters. The soil ground spectrum, measured with dried and ground soil samples, is regarded as a "pure spectrum" that only reflects the soil's chemical composition information [51]. Based on this, a spectral correction model was constructed, which took the pixel spectrum and three sets of soil physical parametric simulated spectra as input and laboratory-measured spectra as training targets. Through multi-order polynomials and various machine-learning algorithms, correction relations between the pixel spectrum and the ground spectrum were established to strip the reflection information attributed to soil physical properties in the pixel spectrum. The multi-order polynomial equation is as follows:

$$R_G = \sum_{i=1}^{n} (a_i R_{SM} + b_i R_{RMSH} + c_i R_{SBW} + d_i R_P)^i + e \tag{1}$$

where $R_G$ is the laboratory-measured soil reflectance, $R_{SM}$ is the soil reflectance simulated based on SM, $R_{RMSH}$ is the soil reflectance simulated based on RMSH, $R_{SBW}$ is the soil reflectance simulated based on SBW, $R_P$ is the soil reflectance of the image pixel, $i$ is the polynomial order, and $a_i$, $b_i$, $c_i$, $d_i$, and $e$ are regression coefficients.

### 2.4. Machine-Learning Models
### 2.4.1. Competitive Adaptive Reweighted Sampling (CARS)

CARS is adopted to extract sensitive bands corresponding to SOM in the hyperspectral data. CARS imitates the "survival of the fittest" principle of Darwin's evolutionary theory. Through adaptive weighted sampling, it screens out the wavelengths with large absolute coefficients of the PLS model and removes the wavelengths with small weights, thus obtaining many subsets of wavelength variables. Next, the subset of wavelengths with the lowest root-mean-square error is selected via cross-validation as the optimal subset [42,52].

### 2.4.2. eXtreme Gradient Boosting (XG-Boost)

XG-Boost is an ensemble-learning model based on the boosting strategy, which combines several CART trees into a strong learner. As an ensemble algorithm framework, it supports the parallel gradient lifting of the base learner, thus greatly improving the model-training speed. The Newton method is used to solve the extreme value of the loss function, which is expanded to the second order using the Taylor formula. The loss function is optimized with the first-order gradient function and second-order gradient function to reduce the model complexity [53]. Simultaneously, the probability for over-fitting is reduced through regularization, significantly improving the model's generalization ability.

### 2.4.3. Model Validation

In this study, the coefficient of determination ($R^2$), root-mean-square error ($RMSE$), and residual predictive deviation ($RPD$) were selected as evaluation indices, as expressed below:

$$R^2 = 1 - \sum_{i=1}^{n} (y_i - Y_i)^2 \Big/ \sum_{i=1}^{n} (y_i - \overline{y})^2 \tag{2}$$

$$RMSE = \sqrt{\frac{1}{n} \sum_{i=1}^{n} (y_i - Y_i)^2} \tag{3}$$

$$RPD = SD / RMSE \tag{4}$$

where $n$ is the number of samples, $y_i$ is the measured value of sample $i$, $Y_i$ is the predicted value of sample $i$, and $\overline{y}$ is the average value of the measurements.

## 3. Results

### 3.1. Descriptive Statistics of Soil Parameters

The basic statistical analysis results of the soil parameters are listed in Table 2. At Site 1, the mean SM, RMSH, and SBW values were 0.25 $cm^3/cm^3$, 2.49 cm, and 0.98 $g/cm^3$, respectively, with standard deviations (SDs) of 0.08 $cm^3/cm^3$, 0.77 cm, and 0.15 $g/cm^3$. The SOM content varied significantly from 25.84 to 75.97 g/kg, with an SD of 10.51 g/kg and a coefficient of variation (CV) of 24.30%. Site 2 had significantly different soil properties from Site 1. The average SM, RMSH, and SBW were 0.37 $cm^3/cm^3$, 3.65 cm, and 1.13 $g/cm^3$, respectively, which were significantly higher than those at Site 1 and had stronger variabilities. The SOM content at Site 2 ranged from 27.40 to 72.97 g/kg, averaging 41.57 g/kg, which was lower than that at Site 1.

**Table 2.** Descriptive statistics of soil parameters and SOM content.

| Dataset | Unit | Site 1 | | | | | Site 2 | | | | |
|---------|------|--------|--------|--------|--------|--------|--------|--------|--------|--------|--------|
| | | Min. | Max. | Mean | SD | CV% | Min. | Max. | Mean | SD | CV% |
| SM | $cm^3/cm^3$ | 0.14 | 0.47 | 0.25 | 0.08 | 31.99 | 0.21 | 0.63 | 0.37 | 0.14 | 37.93 |
| RMSH | cm | 1.32 | 4.99 | 2.49 | 0.77 | 30.92 | 2.04 | 5.78 | 3.65 | 1.34 | 36.71 |
| SBW | $g/cm^3$ | 0.71 | 1.41 | 0.98 | 0.15 | 15.31 | 0.85 | 1.51 | 1.13 | 0.18 | 15.92 |
| SOM | g/kg | 25.84 | 75.97 | 43.25 | 10.51 | 24.30 | 27.40 | 72.97 | 41.57 | 10.28 | 24.72 |

### 3.2. Effect of Soil Physical Properties on Soil Spectra

To verify the reliability of the ZY1-02D hyperspectral image, the soil pixel spectrum was compared with the soil ground spectrum (Figure 4). Although the soil pixel spectrum has a similar shape to that of the soil ground spectrum, it has some noise and a relatively low smoothness, especially in the VNIR wavelength range. In addition, the spectral reflectance in the soil pixels was slightly lower than that measured in the laboratory. The Spearman correlation coefficients (SCCs) and Pearson correlation coefficients (PCCs) between the pixel reflectance and ground reflectance were calculated. The results showed that the PCCs were below 0.5 at most wavelengths, while the SCCs in the 480 to 680 nm and 2000 to 2500 nm wavelength ranges were basically greater than 0.5, indicating a possible nonlinear relationship. To reveal the factors affecting the pixel spectrum, the differences in the soil reflectance between different physical property gradients were compared. With the increase in SM, the soil spectral reflectance decreased significantly, especially in the 500 to 1300 nm and 1450 to 1700 nm wavelength ranges (Figure 5). The soil spectral reflectance decreased relatively slightly with the increase in SBW. The effect of the RMSH on the soil spectrum was the most significant, and the reflectance decreased significantly with the increase in RMSH. In summary, the coupling effect of the SM, SBW, and RMSH on the spectrum is an important reason for the deviation in the two sets of spectral data, which severely limits the acquisition of the soil's "pure spectrum" by the imaging spectrometer. Therefore, it is necessary to separate the soil physical and chemical information in the pixel spectral data and improve the SOM prediction accuracy of hyperspectral remote sensing.

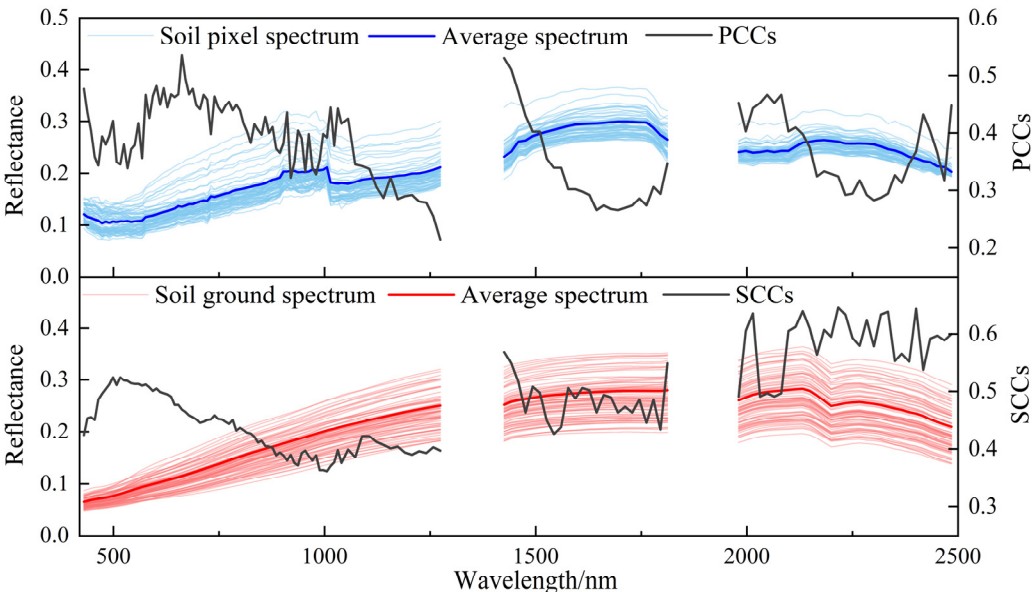

**Figure 4.** Imaging spectrum, laboratory spectrum, and their correlation coefficients.

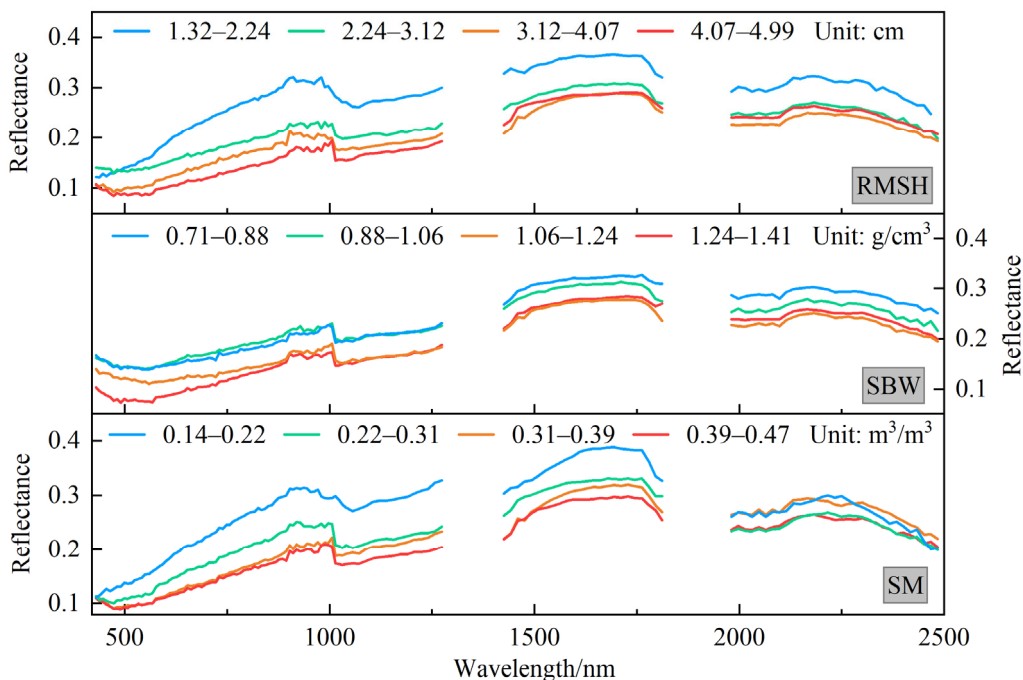

**Figure 5.** Spectral characteristics of soils with different physical properties.

### 3.3. Empirical Relationship between Satellite Hyperspectral Image and Soil Physical Properties

The empirical coefficients were regressed based on the field data and soil pixel spectrum to determine the relationship of the soil reflectance with SM, RMSH, and SBW (Figure 6). Among the fitting equations between the SM and soil reflectance, the exponential equation has the best fitting effect. Except for the 2000 to 2500 nm wavelength range, the fitting results were good, with $R^2$ ranging from 0.49 to 0.68. In the 2000 to 2500 nm wavelength range, the fitting between the SM and reflectance was not good, possibly because of the effect of the absorption of clay minerals on the spectral characteristics, caused by the hydroxyl groups in the soil. With higher clay mineral contents, the water retention capacity of the soil was greater. According to the fitting results between the SBW and soil reflectance, the exponential equation fitted the best in the 450 to 1800 nm wavelength range, with $R^2$ ranging from 0.50 to 0.69, while the power exponential equation fitted the best in

the 2000 to 2500 nm wavelength range. In terms of the entire wavelength range, RMSH had the strongest fitting relationship with the soil reflectance among the three groups of soil physical parameters, implying its most significant effect on the soil spectrum. Among the four equations, the logarithmic equation had the best fitting effect, with $R^2$ ranging from 0.55 to 0.69. In general, the best empirical relationship of the soil reflectance with SM is exponential, that with RMSH is logarithmic, and that with SBW is exponential in the 450 to 1800 nm wavelength range and power exponential in the 2000 to 2500 nm wavelength range. Three sets of soil reflectance data were simulated based on the empirical relationship between soil physical parameters and soil spectra (Figure 7). The soil reflectance simulated based on SM showed an almost uniform trend between 2000 and 2500 nm, implying that the effect of the SM on the reflectance in this wavelength range was suppressed by other factors, resulting in insignificant spectral features. Other than that, the remaining simulated soil spectra showed significant differences.

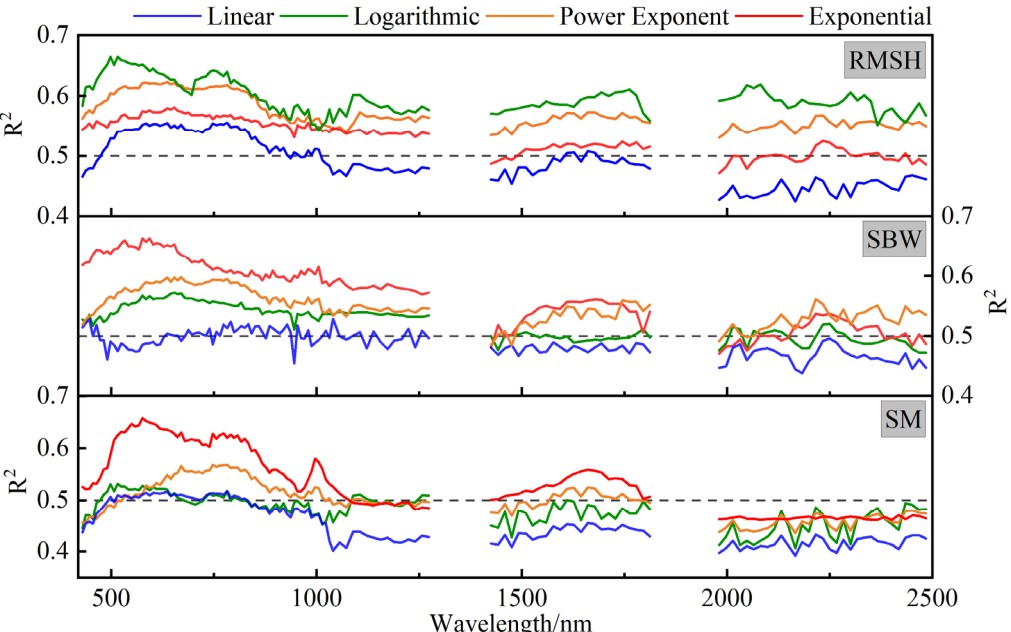

**Figure 6.** $R^2$ for fitting soil physical parameters to soil pixel spectrum based on multiple parameter estimation models.

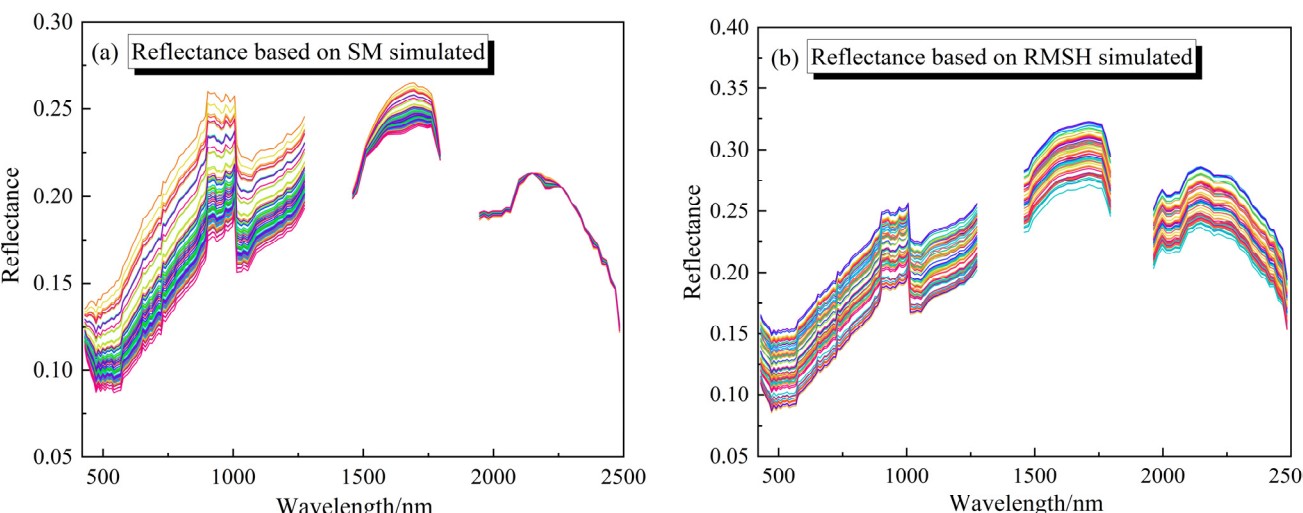

**Figure 7.** *Cont.*

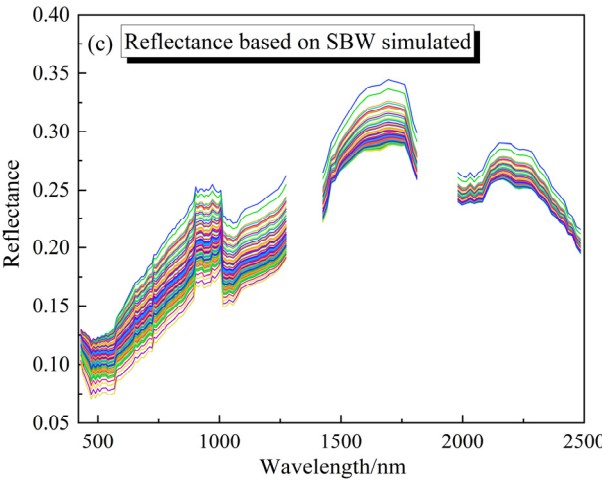

**Figure 7.** Soil spectrum simulated through empirical equations based on SM (**a**), RMSH (**b**), and SBW (**c**).

### 3.4. Modeling of Soil Spectral Correction

Empirical coefficient models and machine-learning models were employed to establish the correction relationship between the soil pixel spectrum and soil "pure spectrum". The multi-order polynomial-based soil spectral correction model showed improved accuracy with the increasing order, and its *RPD* and *RMSE* were optimal in all the bands at the fourth order (Figure 8). When the order reached the fifth order, $R^2$ still maintained an increasing trend, but *RPD* decreased significantly, indicating that the model was over-fitting. The *RMSE* of the fourth-order polynomial model was less than 0.04, and the *RPD* was above 1.5, indicating good correction accuracy.

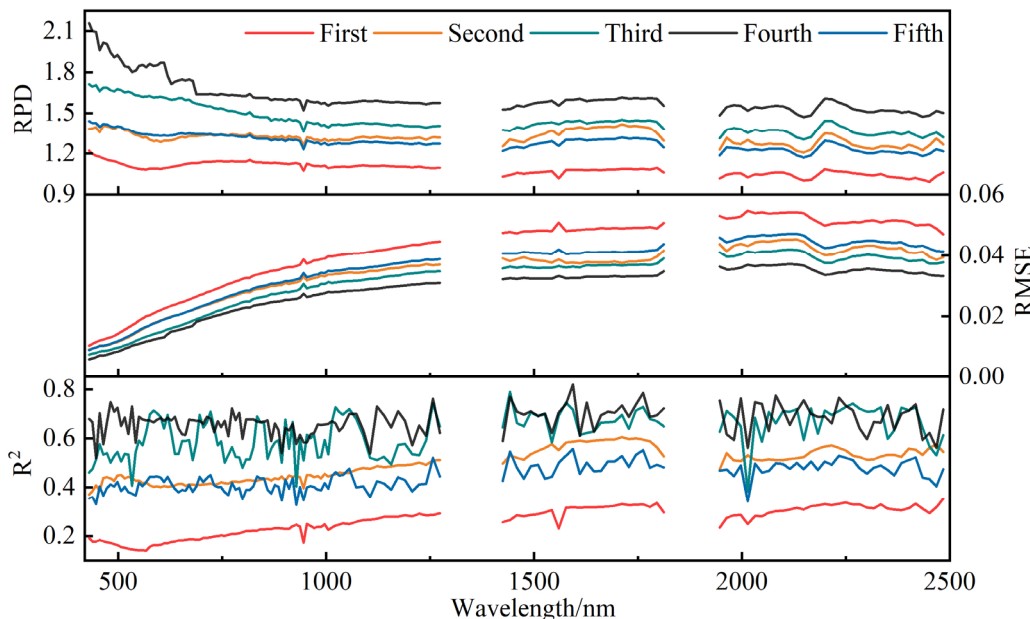

**Figure 8.** Soil spectral correction accuracy based on multi-order polynomial regression.

Four machine-learning algorithms, namely, support vector machine regression (SVR), an extreme learning machine (ELM), a backpropagation neural network (BPNN), and XG-Boost, were used to construct soil spectral correction models in the same way. The best soil spectral correction model was determined by comparing the mapping ability of the different machine-learning algorithms to the coupling relationship between multiple soil spectra (Figure 9). The correction results showed that the four machine-learning algorithms differed significantly in the soil spectral correction accuracy. The accuracy fluctuations

around the 1000 nm wavelength may be caused by other noise in the spectral data. Other than that, all the soil spectral correction results were good, and the overall accuracy was high relative to that of the polynomial-based models. As a representative algorithm of the ensemble-learning model, XG-Boost achieved the best spectral correction results, with $R^2$ above 0.6 and $RPD$ above 1.6 for all the bands. The second best was ELM, while SVM and BPNN performed poorly.

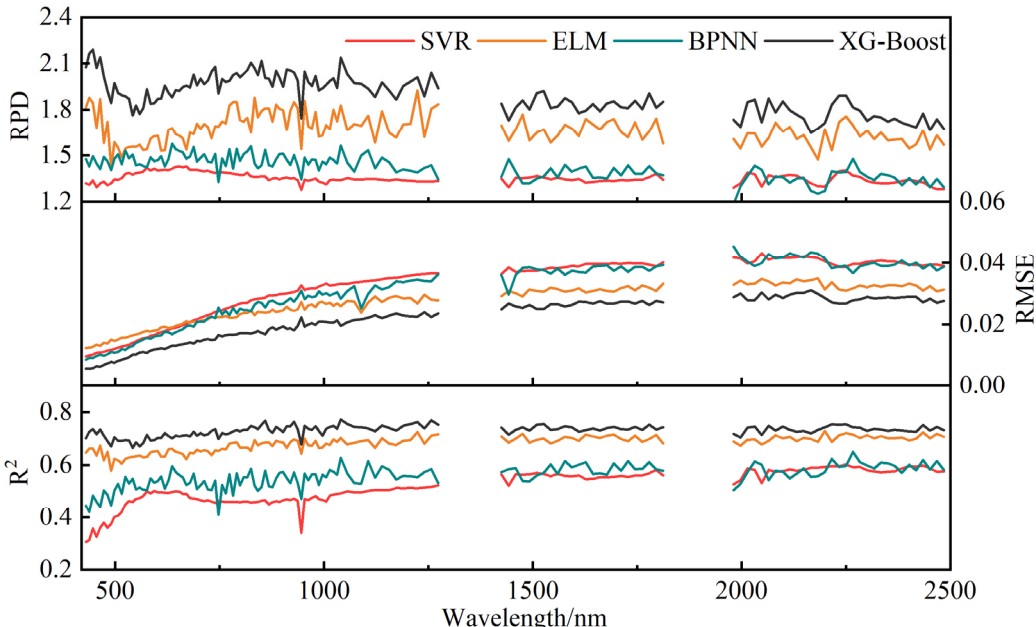

**Figure 9.** Soil spectral correction accuracy based on machine-learning models.

The soil spectral correction accuracy based on the fourth-order-polynomial model and XG-Boost model was high, and these two sets of spectral data were selected for in-depth analysis. The results showed that the corrected spectrum based on XG-Boost was smoother than the corrected spectrum based on the fourth-order polynomial and fit the shape of the soil's "pure spectrum" more closely (Figure 10). The calculated correlation coefficients between the soil's "pure spectrum" and the corrected spectrum showed that the PCCs at most wavelengths were above 0.8, which was greatly improved compared with the correlation between the soil's "pure spectrum" and the original pixel spectrum. Therefore, after the spectral correction, the spectral response, induced by soil physical properties, in the soil pixel spectrum was alleviated, and the proportion of the information on the soil's chemical-composition-response signals in the image spectral data was significantly increased. In general, the correction results of the XG-Boost model are slightly better than those of the fourth-order-polynomial model. However, the accuracy improvement effect of these methods on hyperspectral SOM prediction needs further analysis through modeling.

### 3.5. SOM Content Prediction Accuracy

Four types of soil spectral data, namely, a pixel spectrum, a fourth-order-polynomial-corrected spectrum, an XG-Boost-corrected spectrum, and a ground-based spectrum, were used to establish the SOM content prediction models. To reduce the data dimensionality and improve the computational efficiency of the model, the spectral data were first subjected to feature extraction. Pearson's correlation coefficient threshold was used to determine the sensitive bands of the SOM. The correlation coefficient distribution between the four sets of soil spectral data and SOM content showed a relatively consistent trend. Specifically, the correlation coefficient decreased with the increasing wavelength before the 800 nm wavelength and increased after the 800 nm wavelength (Figure 11a). The bands with absolute correlation coefficients above 0.5 were selected as the sensitive bands of the SOM.

The sensitive spectral bands corresponding to the SOM in the four spectral datasets of the pixel spectrum, fourth-order-polynomial-corrected spectrum, XG-Boost-corrected spectrum, and ground-based spectrum were concentrated in wavelength ranges from 628 to 1023 nm, from 524 to 1223 nm, from 542 to 1560 nm, and from 550 to 1762 nm, respectively. CARS was adopted to further extract the optimal subset of features containing the least-redundant information in the sensitive bands. The optimal number of CARS iterations was determined by the RMSECV of the multiple regression (Figure 11b). The bands listed in Table 3 are the spectral bands selected by CARS for SOM inversion modeling and validation analysis.

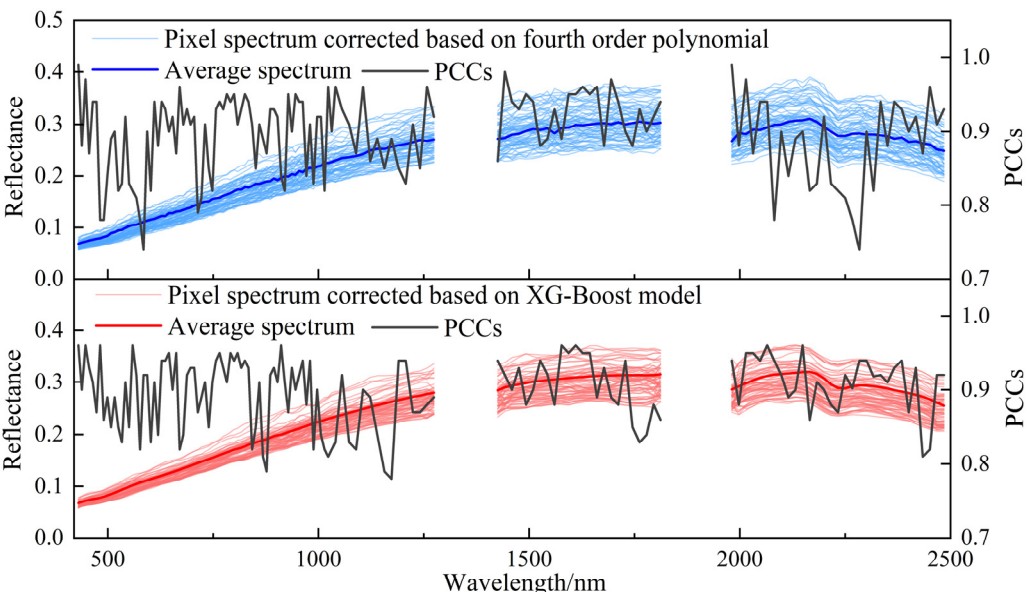

**Figure 10.** Soil pixel spectra corrected with the XG-Boost model and the fourth-order-polynomial model and the correlation coefficient between the corrected spectrum and the laboratory spectrum.

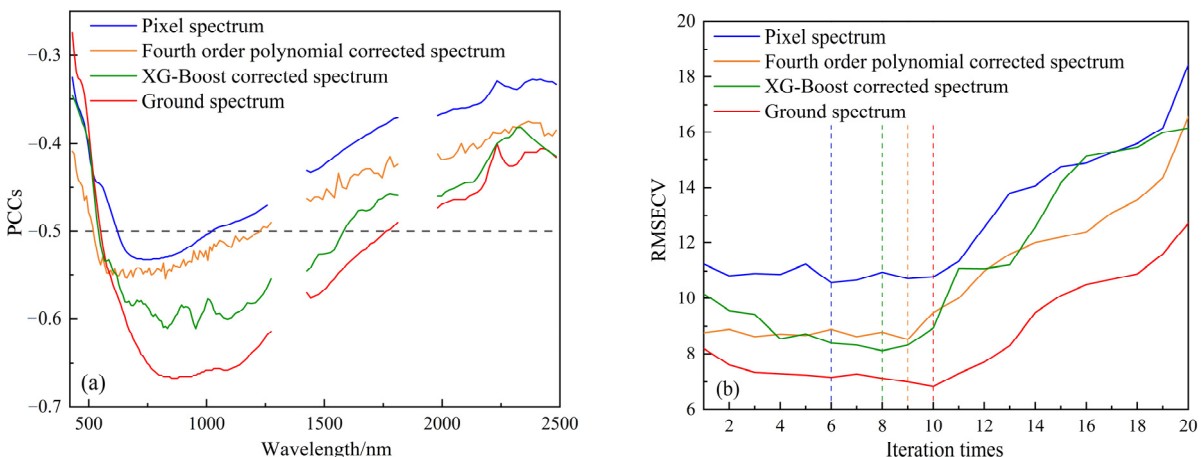

**Figure 11.** (**a**) Pearson's correlation coefficients between SOM contents and spectral reflectance of each band. (**b**) *RMSECV* (unit: g/kg) of multiple regression with different numbers of CARS iterations.

The four sets of spectral bands selected through CARS and the SOM contents were used as the input data of the model. The XG-Boost algorithm is used to construct the SOM prediction model. The results indicated that these two spectral correction methods significantly improved the SOM prediction accuracy based on the original pixel spectrum. Among all the SOM prediction results, the ground spectral data had the highest prediction accuracy, with $R_P^2$, *RMSEP*, and *RPD* values of 0.79, 4.89 g/kg, and 1.97, respectively (Figure 12). The prediction accuracy evaluated based on $R_P^2$ was 0.64 when the original

pixel spectral dataset was used as model input data. Adopting the soil spectral correction strategy based on fourth-order polynomials increased the prediction accuracy ($R_P^2$) by 0.05, decreased *RMSEP* by 2.28 g/kg, and increased *RPD* by 0.38. The soil spectral correction strategy based on the XG-Boost model had a greater SOM prediction accuracy improvement, with an $R_P^2$ increase of 0.12, an *RMSEP* decrease of 3.10 g/kg, and an *RPD* increase of 0.59. The SOM prediction accuracy with the corrected spectrum came close to that with the ground spectrum, implying that the spectral correction model can effectively improve the hyperspectral SOM prediction accuracy.

**Table 3.** Features' band statistics selected based on CARS.

| Spectral Correction Method | Wavelength (Unit: um) | Total |
|---|---|---|
| Pixel spectrum | 0.67; 0.68; 0.70; 0.72; 0.74; 0.77; 0.79; 0.84; 0.87; 0.90; 0.93 | 11 |
| Fourth-order-polynomial-corrected spectrum | 0.55; 0.60; 0.62; 0.68; 0.73; 0.76; 0.78; 0.82; 0.85; 0.87; 0.91; 0.96; 0.99; 1.07 | 14 |
| XG-Boost-corrected spectrum | 0.55; 0.62; 0.64; 0.69; 0.73; 0.77; 0.81; 0.83; 0.87; 0.89; 0.92; 0.94; 0.99; 1.05; 1.17 | 15 |
| Ground-based spectrum | 060; 0.63; 0.67; 0.70; 0.73; 0.77; 0.81; 0.85; 0.87; 0.90; 0.91; 0.96; 0.99; 1.03; 1.08; 1.22 | 16 |

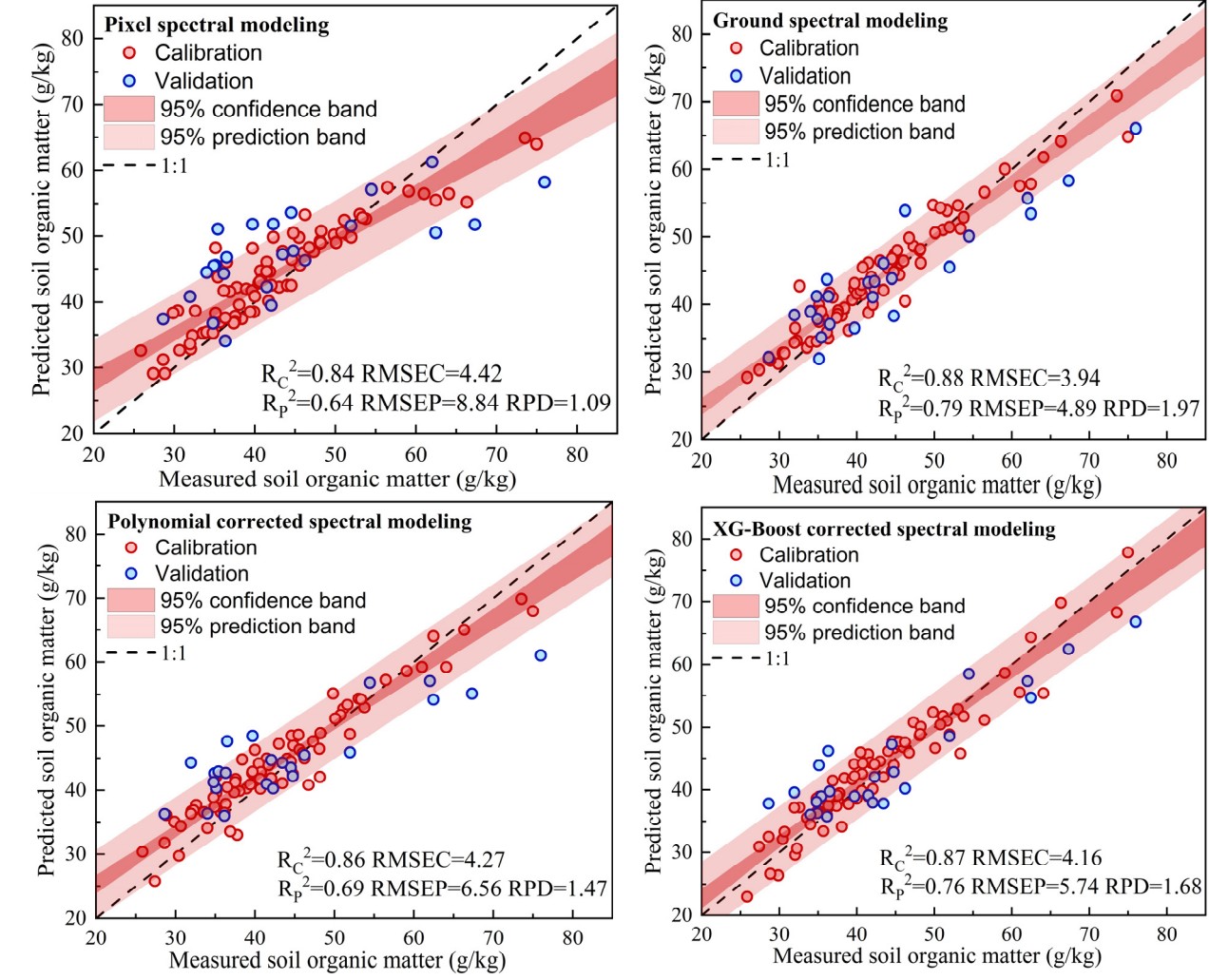

**Figure 12.** Scatter plots of predicted and measured SOM contents based on the four spectral datasets.

## 4. Discussion

### 4.1. The Spatiotemporal Transferability of the Soil Spectral Correction Model and SOM Prediction Model

The soil pixel spectral correction methods based on empirical coefficient and machine-learning models provided new strategies for improving the SOM prediction accuracy based on hyperspectral images. The soil spectral correction method based on XG-Boost significantly affected the SOM prediction accuracy improvement, but its correction process and principles were difficult to express mathematically. Despite its weak SOM prediction accuracy improvement, the soil spectral correction method based on the fourth-order-polynomial model expressed the improved method with the coefficient equation, which was more conducive to its promotion. The high transferability of the soil spectral correction method is a key prerequisite for constructing an SOM prediction model with strong generalization ability [54]. To verify their spatiotemporal transferability, 40 groups of Site 2 soil pixel spectra and ground experimental data were imported into the two spectral correction models. The spectral correction results showed that the corrected soil spectra were very consistent with the shape of the soil's "pure spectra" (Figure 13). Compared with the soil spectrum after the fourth-order-polynomial correction, the soil spectrum corrected with the XG-Boost model is smoother. According to the accuracy of the model migration test results, the soil spectral correction model based on XG-Boost showed better migration performance, with *RPD* above 1.4 for all the bands. The machine-learning algorithms represented by XG-Boost were much better than the coefficient models in terms of the calculation ability to establish the coupling relationship between multiple soil spectra and its adaptability to new data. The reason is that this ensemble-learning model comprehensively utilizes all the eigenvalues of each soil sample point and continuously adjusts the weight of the tree through iteration to explore the optimal solution of the coupling relationship between the soil physical properties and pixel spectrum [43,55].

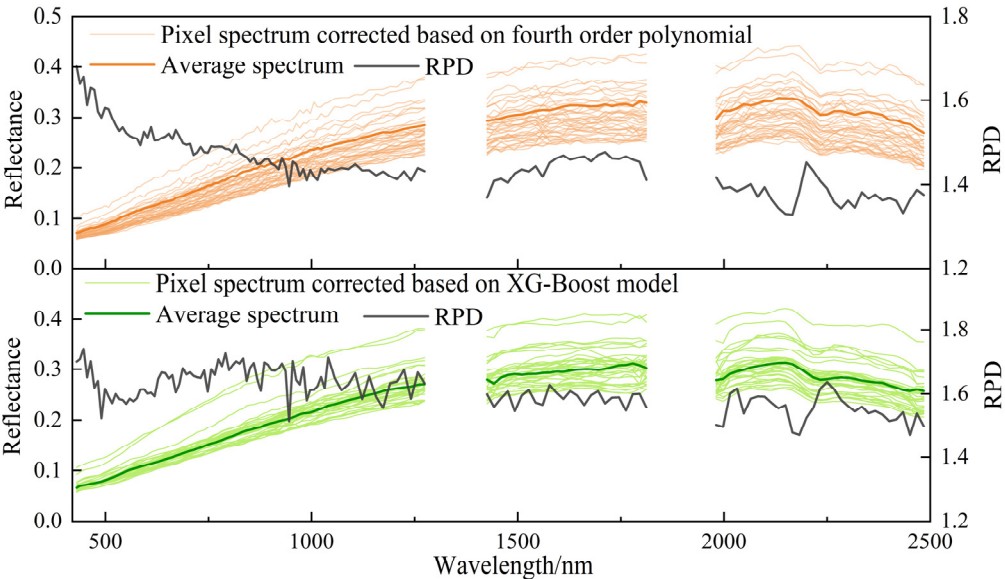

**Figure 13.** Soil pixel spectra corrected with the XG-Boost and fourth-order-polynomial models and the *RPD* of two correction models.

The poor transferability of SOM remote-sensing prediction models is mainly attributed to their poor applicability to different spatiotemporal spectral data [56,57]. To evaluate the improvement effect of the soil spectral correction methods on the spatiotemporal transferability of SOM prediction models, the SOM prediction model established using Site 1 data was used to predict the SOM content at Site 2. The SOM prediction model based on ground spectra exhibited the best transferability, with *RMSEP* only increasing by 0.39 g/kg (Figure 14). However, the transferability of SOM prediction models based on

the original pixel spectrum is extremely poor, as surface physical property changes cause spectral reflectance deviations. After the transferability verification, *RMSEP* increased by 8.05 g/kg, while RDP decreased by 44.04%. Adopting the two soil spectral correction strategies significantly improved the transferability of the prediction model, with an *RPD* of over 1.4 for the model transferability validation. The SOM prediction model based on the XG-Boost-corrected spectrum had greater transferability. Compared with the model transferability validation based on the pixel spectrum, *RMSEP* was reduced by 60.27%, and *RPD* was increased by 150.82%. These findings proved the effectiveness of the soil spectral correction methods and the feasibility of the corrected satellite hyperspectral data in SOM content predictions. The SOM prediction model based on corrected satellite hyperspectral data can be used even at two sites with different soil types, soil physical properties, SOM contents, and spatiotemporal features. The core of this soil spectral correction method is to comprehensively consider the coupling effect of various soil properties on spectral reflectance to restore the true spectral characteristics of the research target. For different research objectives, the main factors affecting the spectral response of the target can be analyzed according to the actual environment and imaging conditions [35]. Therefore, the proposed method is not limited to SOM predictions and can provide valuable insights for soil property predictions based on satellite hyperspectral data.

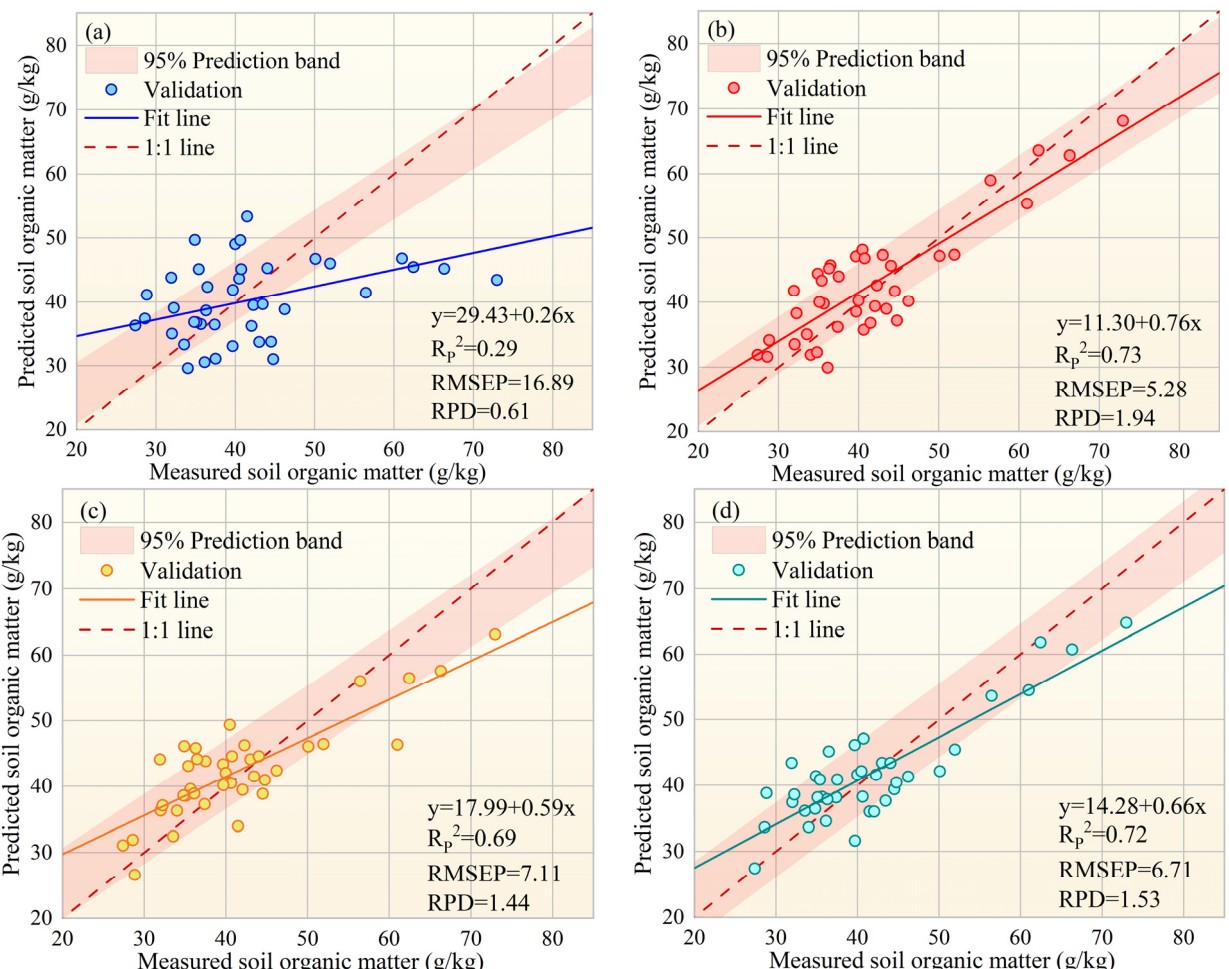

**Figure 14.** Scatter plots of the measured and predicted SOM contents based on (**a**) original pixel spectrum, (**b**) ground spectrum, (**c**) fourth-order-polynomial-corrected spectrum, and (**d**) XG-Boost-corrected spectrum with Site 2 data, using the XG-Boost model established using Site 1 data.

### 4.2. Contribution of Soil Physical Properties to SOM Content Prediction Bias

The soil spectral correction method proved to greatly improve the SOM content prediction accuracy and spatiotemporal transferability of the pixel spectrum. In other words, soil properties (SM, RMSH, and SBW) may be the main factors leading to errors in SOM estimation based on original pixel spectra. This section investigated the error dependence of the original pixel spectrum and two sets of corrected spectral data on SM, RMSH, and SBW, and their contribution to the SOM content prediction bias was estimated through the stepwise regression method. The results showed that the cumulative deviation contribution rate of these three soil properties to the SOM prediction results based on the original pixel spectrum was over 70% at both sites (Figure 15). Thus, soil physical properties are the main error source of SOM predictions [50]. The contribution of the SM to the SOM bias was the highest, followed by RMSH and SBW. This is related to the most significant response of the pixel spectrum to SM within the sensitive wavelength range of the SOM and possibly the stronger spatial heterogeneity of the SM compared to RMSH and SBW. Adopting the spectral correction strategy significantly reduced the bias of the SM, RMSH, and SBW in SOM predictions. The soil spectral correction based on XG-Boost more significantly reduced the SOM prediction bias caused by soil physical properties than the soil spectral correction based on fourth-order polynomials. Despite the increased relative contribution of random errors, the total bias in terms of the accuracy of the predicted results was significantly reduced. Therefore, the spectral correction strategy did not introduce more error sources but only improved the relative contribution of other error factors to the SOM prediction bias, such as hyperspectral-image-processing uncertainty and field data acquisition uncertainty [58,59]. Judging from the relative contribution rate of the soil physical properties to the SOM prediction deviation before and after the spectral correction, the contribution rate of the SM to the SOM prediction bias decreased the most, by over 10%, followed by RMSH. The spectral correction method based on polynomials and XG-Boost reduced the average relative contribution rate of the RMSH at the two sites by 10% and 14.5%, respectively. Although the declined contribution rate of the SBW to the SOM prediction bias was the smallest, the reduction exceeded 6%. By comparing the improvement effects of different input variables on the SOM prediction bias, the improvement effects of soil physical properties on the SOM content prediction accuracy ranked as SM > RMSH > SBW, consistent with the order of the spatial heterogeneity of these three soil physical properties and the order of the sensitivity of soil spectra to them in the VNIR range. Thus, soil physical properties with strong spatial heterogeneity and a sensitive spectral response should be prioritized in soil spectral correction.

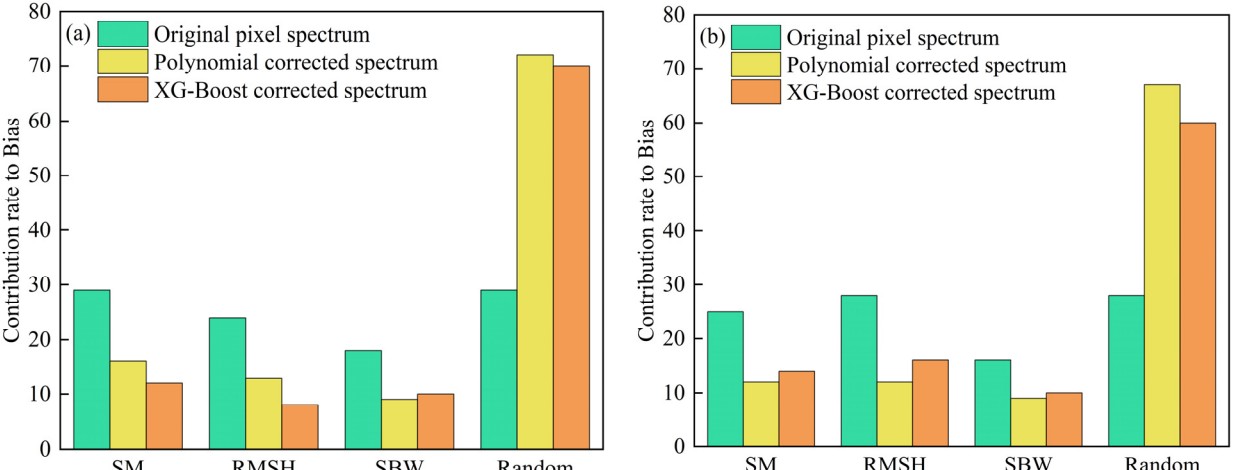

**Figure 15.** Contribution rates of soil properties (SM, RMSH, and SBW) to the estimated SOM bias at Site 1 (**a**) and Site 2 (**b**). "Random" denotes the part that these three variables cannot explain.

### 4.3. The Potential and Limitations of the Soil Spectral Correction Model

As soil spectral correction methods are designed to address the coupled effects of surface physical properties on hyperspectral images, they are suitable for the remote-sensing image processing of various soil chemical composition predictions. Such methods suppress the sensitivity of spectral data to SM, RMSH, and SBW and reduce the possibility of SOM prediction results falling into the local optimum. Another advantage is that determining the empirical relationship of SM, RMSH, and SBW with the hyperspectral soil reflectance of ZY1-02D satellite spectra improves the generalization ability and application efficiency of the method. In addition, this study has great potential application value and is expected to realize the combined application of optical and radar remote sensing in the estimation of soil physical and chemical properties. Although the soil spectral correction model can restore most soil "pure spectrum" characteristics, some uncertainties may remain. The applicability of the proposed method to airborne hyperspectral sensors or other hyperspectral satellites requires evaluation with more data. In addition, this experiment only considered the influence of soil properties within 5 cm of the surface layer on the spectrum, while the spectrum may have different sensing depths for different soil properties [60]. Even though the spectrum only directly detects SM changes in shallow soil (from about 0 to 2 cm), this depth also changes under different SBW and RMSH conditions [20,61]. The vertical heterogeneity of the SM and SBW may be the main factor causing soil spectral correction errors. Adopting a hierarchical strategy to establish spectral correction models for soil properties at different depths or assigning different weights to soil physical properties at different depths may maximize the effectiveness of the model within the specified depth and range.

### 4.4. Future Work and Suggested Next Steps

This study, for the first time, used the "pure spectrum" derived from soil spectral correction, considering SM, RMSH, and SBW, for SOM content prediction and confirmed its excellent spatiotemporal transferability. The strategy for alleviating the influence of soil physical properties on spectral coupling may provide a paradigm for remote-sensing-based soil element content predictions in the future. The soil spectral correction of the whole hyperspectral image requires real-time high-resolution soil physical parameter data at the regional scale. The strong sensitivity of synthetic aperture radar remote sensing to soil physical parameters makes it possible to obtain rich information on soil physical parameters in practical applications [62]. Future research may combine the advantages of hyperspectral imaging and radar remote sensing to improve the prediction accuracy of soil physicochemical parameters [63]. Given the type and dimensional heterogeneity of optical and radar data, a new link to combine these two data types was provided. Meanwhile, some interesting new directions have also emerged. Because of the difference in the imaging time between radar and hyperspectral images, the corresponding surface physical properties also change, especially SM, which is highly susceptible to weather. Therefore, combining optical and radar data to eliminate the bias due to the temporal phase may be the optimal strategy to solve this problem. In addition, the depth perception of optical and radar remote sensing for soil properties needs further clarification. As radar sensors have better penetrability than optical sensors, quantifying the vertical heterogeneity of SM and SBW with radar remote sensing could be the key to further improving the soil spectral correction accuracy [64,65]. These strategies reduce errors in large-scale soil physical and chemical parameter surveys to support regional strategic arrangements for sustainable agricultural development.

### 5. Conclusions

This study utilized satellite and ground hyperspectral data and soil physical parameter data to construct two soil spectral correction models based on fourth-order polynomials and XG-Boost, respectively, to alleviate the coupling effect of soil physical properties on the pixel spectrum. The performance of the soil spectral correction models and their influence

on the accuracy and spatiotemporal transferability of the SOM prediction model were evaluated using data from two sites. The main conclusions are as follows. (1) The soil pixel spectral reflectance is nonlinearly related to the soil ground spectral reflectance. The difference in the surface physical properties is the main factor for the deviation in these two spectral data types. RMSH has the most significant effect on the soil pixel spectrum, followed by SM and SBW. (2) The fourth-order-polynomial and XG-Boost models have good soil spectral correction accuracies. The soil spectral correction model based on XG-Boost has higher accuracy and stronger spatiotemporal transferability, as it considers all the features to continuously adjust the weight of the tree and prevent the result from falling into the local optimum. (3) Soil spectral correction significantly alleviates the coupling effect of soil physical properties on soil pixel spectra, effectively improves the accuracy of the SOM prediction model, and, more importantly, greatly enhances the spatiotemporal transferability of the SOM prediction model based on the pixel spectrum. In the future, more accurate SOM prediction results can be obtained by fully considering more soil properties. This work provides a new research paradigm for predicting soil property parameters in other regions.

**Author Contributions:** Conceptualization, N.L. and Y.S.; methodology, R.J.; software, H.Y.; validation, X.Z.; formal analysis, R.J.; investigation, N.L. and R.J.; resources, R.J.; data curation, N.L., R.J. and X.Z.; writing—original draft preparation, R.J. and Y.S.; writing—review and editing, R.J. and Y.S.; visualization, N.L. and Y.S.; supervision, R.J.; project administration, R.J.; funding acquisition, N.L., H.Y. and X.Z. All authors have read and agreed to the published version of the manuscript.

**Funding:** This work was supported by the Jilin Provincial Scientific and Technological Development Program (20230101373JC and 20240303035NC), the 14th Five-Year National Key Research and Development Plan of China (2022YFD1500504), and the Common Application Support Platform for Land Observation Satellites of China's Civil Space Infrastructure Project, China (CASPLOS-CCSI).

**Data Availability Statement:** The data presented in this study are available on request from the corresponding author.

**Conflicts of Interest:** The authors declare no conflicts of interest.

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
