# Peer review of "Improving the Spatiotemporal Transferability of Hyperspectral Remote Sensing for Estimating Soil Organic Matter by Minimizing the Coupling Effect of Soil Physical Properties on the Spectrum: A Case Study in Northeast China"

_agronomy, doi:10.3390/agronomy14051067_

Round 1

Reviewer 1 Report

Comments and Suggestions for Authors

Response to: Improving the spatiotemporal transferability of hyperspectral remote sensing for estimating soil organic matter by minimizing the coupling effect of soil physical properties on the spectrum.

Major comments

Thank you to the authors for the study. The study seems to provide some useful methods or approach for rapid soil organic matter estimation. However, my main concern is that the authors fail to explain why SOM rather than SOC was targeted in this work. Why would the authors consider SOM which is difficult to estimate via rapid proximal or remote sensing techniques over the usual SOC which has already been well-proven to correlate with different spectral features or bands? Moreover, the current study knowledge gap is not clear. What is new about the current study? The authors are advised to explicitly state this gap in the introduction and highlight it in the abstract as well. I also advise the authors to summarise the introduction, it is too long and a bit confusing in some parts.

Specific comments

1. Line 37-39: I am uncertain what this sentence means. The authors should revise it.

2. Line 50-53: How is this sentence related to SOM? Here the authors talk about soil elemental levels which is not at all related to the prior theme.

3. Line 64: What is SM content? The authors only use this prefix but fail to provide the full meaning of it before using it. It was only provided in the abstract, but it is well-needed here in the introduction as well.

4. Line 86: What does the macro scale mean?

5. How were the machine learning models trained? (e.g., Sample splitting, Cross-validation)

6. Why did the authors only test or explore two machine-learning models? I see the selection of the two machine learning methods as arbitrary.

7. Most parts of the study require improving the text to ensure clarity and understandability. I found several sentences and paragraphs confusing and, in most cases, unclear.

Comments on the Quality of English Language

There is a need to revise some parts of the manuscript to improve the clarity and flow. 

Author Response

Response to Reviewer Comments

Dear reviews:

Thank you for your letter and for the reviewers’ comments concerning our manuscript entitled “Improving the spatiotemporal transferability of hyperspectral remote sensing for estimating soil organic matter by minimizing the coupling effect of soil physical properties on the spectrum” (ID: agronomy-2991181). Those comments are all valuable and very helpful for revising and improving our paper, as well as the important guiding significance to our researches. We have studied comments carefully and have made correction which we hope meet with approval. The main corrections in the paper and point-by-point response to the reviewer’s comments are as flowing:

Reviewer: 1
Comments to the Author

Thank you to the authors for the study. The study seems to provide some useful methods or approach for rapid soil organic matter estimation. However, my main concern is that the authors fail to explain why SOM rather than SOC was targeted in this work. Why would the authors consider SOM which is difficult to estimate via rapid proximal or remote sensing techniques over the usual SOC which has already been well-proven to correlate with different spectral features or bands? Moreover, the current study knowledge gap is not clear. What is new about the current study? The authors are advised to explicitly state this gap in the introduction and highlight it in the abstract as well. I also advise the authors to summarize the introduction, it is too long and a bit confusing in some parts. I would like to provide some explanation for your concern that the research objective of this article is SOM rather than SOC

Response:

Thank you for your comment. According to your opinion, we have made a major revision to the introduction and clarified the knowledge gap and the new progress of this study. Previous studies on SOM prediction lack consideration of soil physical properties, ignoring the spectral response of soil physical properties, such as such as soil moisture, soil bulk weight, and surface roughness properties, resulting in poor accuracy and spatiotemporal transferability of SOM prediction models. This study aims to improve the spatiotemporal transferability of the SOM prediction model by alleviating the coupling effect of soil physical properties on the spectra. Based on your suggestion, we have highlighted this point in the introduction and abstract sections. We would like provide some explanation for your concern that the research objective of this article is SOM rather than SOC. SOC is the carbon component of SOM, both of which have good spectral response characteristics. A large number of studies have shown that both SOC and SOM can be estimated by fast proximal or remote sensing technology. As an important dyeing material of soil, SOM can significantly absorb the incident light in the visible light band or even the short-wave infrared band, and there is usually a significant negative correlation between its content and soil spectral reflectance. Compared with SOC, SOM is a more important evaluation index of soil nutrient quality, because it can reveal the soil fertility. We believe that it is more meaningful to accurately monitor SOM in the agricultural field, so we take SOM as the research goal. For the reasons for choosing SOM, we have made a major revision to the first paragraph of the introduction. Thank you very much for your professional review. Your valuable comments have helped us significantly improve the quality of the article.

Response of specific comments

  1. Line 37-39: I am uncertain what this sentence means. The authors should revise it.

Response: Thank you for your comment. According to your opinion, we highlighted the importance of accurate monitoring of soil conditions to ensure sustainable agricultural development and global food security in lines 38-52, and clearly pointed out the important role of organic matter in the soil. Thank you very much for your opinion, which makes the research objectives and significance of the revised manuscript clearer.

  1. Line 50-53: How is this sentence related to SOM? Here the authors talk about soil elemental levels which is not at all related to the prior theme.

Response: Thank you for your comment. According to your opinion, we deleted this sentence. In this part, we originally wanted to emphasize that with the rapid growth of remote sensing data and the urgent need for large-scale soil surveys, the research on SOM content prediction based on remote sensing has gradually shifted from building high-precision prediction models to establishing prediction models with strong spatial and temporal transfer. After seriously considering your comments, we believe that it is too early to put forward this view here. Therefore, we change the expression of lines 61-64 of the manuscript to that the image spectrum is affected by soil physical parameters, which is the key to limiting the accuracy of the SOM prediction model.

  1. Line 64: What is SM content? The authors only use this prefix but fail to provide the full meaning of it before using it. It was only provided in the abstract, but it is well-needed here in the introduction as well.

Response: Thank you for your comment. According to your opinion, we defined the abbreviation 'SM' when we first mentioned the word 'soil moisture' in line 66.

  1. Line 86: What does the macro scale mean?

Response: Thank you for your comment. According to your opinion, this unprofessional statement has been deleted, and similar errors in the full text have been checked and modified.

  1. How were the machine learning models trained? (e.g., Sample splitting, Cross-validation)

Response: Thank you for your comment. According to your opinion, we supplement the division of training samples and validation samples in lines 189-192. A total of 104 surface soil samples were collected from Site 1 between October 29-30, 2022. Among them, 80 soil samples were used as the training set for the models, and the remaining 24 samples were used as the validation set. Between April 14-15, 2023, 40 surface soil samples were collected from Site 2 for testing the spatiotemporal transferability of the models.

  1. Why did the authors only test or explore two machine-learning models? I see the selection of the two machine learning methods as arbitrary.

Response: Thank you for your comment. According to your comment, we would like to explain that we used multi-order polynomial regression and a variety of machine learning algorithms to establish soil spectral correction models to determine the optimal modeling method. We tested the multi-order polynomial regression accuracy of 1-5 orders and the spectral correction accuracy of various machine learning algorithms such as SVM, ELM, BPNN and XG-Boost. Based on the soil spectral correction accuracy, the soil spectral correction results of the fourth-order polynomial model and XG-Boost model were selected for further analysis. Thank you very much for your opinion. We elaborate on the comparison of the accuracy of various machine learning algorithms in lines 358-370.

  1. Most parts of the study require improving the text to ensure clarity and understandability. I found several sentences and paragraphs confusing and, in most cases, unclear.

Response: Thank you for your comment. According to your comment, we have reviewed and corrected the misleading descriptions in the manuscript, and polished the language. Especially in the introduction, the logical flow is reorganized and the sentences and paragraphs that are difficult to understand are modified. The main modifications are as follows: lines 38-47, 54-61, 84-88, 201-205, 238-243, and 456-460.

In addition to our point-by-point replies to the reviewers’ comments above, we also checked and modified the language of the full text. We did our best to meet the standards of required editorial corrections and have made all changes easily identifiable. We hope that our revised manuscript meets your requirements. If any further action is needed, please let us know immediately. We look forward to hearing back from you.

Thank you and best regards.

Yours sincerely,

The Authors

Reviewer 2 Report

Comments and Suggestions for Authors

The manuscript addresses an important topic, establishing the SOM content prediction model with high accuracy and strong spatiotemporal transferability for a soil spectral correction to alleviate the coupling effect of surface physical properties on the soil pixel spectrum. However, several major comments require to be addressed

1)      What is the gap in this research area exactly?

2)      The gap research would shape the objective as the objective is not clear how deep the model can be accurately relevant to hose in the literature

3)      The outcome is only relevant to one case study, china

4)      The suggestion of changing the title to include something relevant to the case study

Other minor comments are

1)      Figure 1 is not clear and the flowchart needs to be addressed adequately

2)      Is not clear what are the output in physical soil parameters that would be addressed

3)      What is the relation between dedicating the soil's physical properties and sustainability along with the carbon existing in the soil while managing dedication for better agriculture is not clear the relevant topics together

Author Response

Response to Reviewer Comments

Dear reviews:

Thank you for your letter and for the reviewers’ comments concerning our manuscript entitled “Improving the spatiotemporal transferability of hyperspectral remote sensing for estimating soil organic matter by minimizing the coupling effect of soil physical properties on the spectrum” (ID: agronomy-2991181). Those comments are all valuable and very helpful for revising and improving our paper, as well as the important guiding significance to our researches. We have studied comments carefully and have made correction which we hope meet with approval. The main corrections in the paper and point-by-point response to the reviewer’s comments are as flowing:

Reviewer: 2
Response of specific comments

  1. What is the gap in this research area exactly?

Response: Thank you for your comment. According to your opinion, we have made a major revision to the introduction and clarified the knowledge gap and the new progress of this study. Previous studies on SOM prediction lack consideration of soil physical properties, ignoring the spectral response of soil physical properties, such as such as soil moisture, soil bulk weight, and surface roughness properties, resulting in poor accuracy and spatiotemporal transferability of SOM prediction models. This study aims to improve the spatiotemporal transferability of the SOM prediction model by alleviating the coupling effect of soil physical properties on the spectra. According to your opinion, we have highlighted this point in the lines 61-69.

  1. The gap research would shape the objective as the objective is not clear how deep the model can be accurately relevant to hose in the literature.

Response: Thank you for your comment. According to your opinion, we have made clear the objective of this study. The existing studies on SOM prediction often lack the consideration of soil surface physical properties, resulting in poor generalization ability of the prediction models. This study combined satellite hyperspectral images with soil physical variables (soil moisture, soil surface roughness, and soil bulk density) to construct a soil spectral correction strategy based on the information decomposition method to alleviate the influence of soil physical properties on the hyperspectral images, thereby improving the accuracy and spatiotemporal transferability of SOM prediction models. Based on your suggestion, we have highlighted this point in the lines 104-113.

  1. The outcome is only relevant to one case study, China.

Response: Thank you for your comment. As you commented, our research is based on the black soil arable land in Northeast China. Due to the lack of soil data in other countries and regions, we cannot verify the effectiveness of this method in other regions. However, through the migration verification of SOM prediction model, we have confirmed that our proposed soil spectral correction method can effectively improve the accuracy and generalization performance of SOM prediction model. Thank you very much for your valuable advice. In the future, we will consider more black soil cultivated land in the same dimension to carry out the migration test of SOM prediction model.

  1. The suggestion of changing the title to include something relevant to the case study

Response: Thank you for your comment. According to your opinion, we have revised the title to 'Improving The Spatiotemporal Transferability of Hyperspectral Remote Sensing for Estimating Soil Organic Matter by Minimizing The Coupling Effect of Soil Physical Properties on The Spectrum -A Case Study in Northeast China'.

  1. Figure 1 is not clear and the flowchart needs to be addressed adequately.

Response: Thank you for your comment. According to your opinion, we modified the figure 1. Combined with your sixth opinion, we focus on modifying the role of soil physical parameters in the spectral analysis section and the output of soil physical parameters in the model.

  1. Is not clear what are the output in physical soil parameters that would be addressed.

Response: Thank you for your comment. According to your opinion, we explained the output results of soil physical parameters in lines 148-152. The main purpose of soil physical parameters is to determine the impact of soil physical properties on spectra, and to conduct forward spectral simulation to obtain simulated spectral data based on soil physical properties. Firstly, parameter estimation equations were used to establish empirical relationships between satellite hyperspectral data and the three main soil physical parameters SM, RMSH, and SBW. Then, the spectral data of soil physical properties were obtained by spectral forward simulation using soil physical parameters.

  1. What is the relation between dedicating the soil's physical properties and sustainability along with the carbon existing in the soil while managing dedication for better agriculture is not clear the relevant topics together.

Response: Thank you for your comment. According to your opinion, we have modified the first paragraph of the introduction and clarified the theme of this study. According to your comment, we would like to explain that the object of this study is to explore SOM prediction methods with spatial and temporal mobility to accurately understand SOM content, so as to provide basic data for promoting sustainable agricultural development, improving soil carbon sequestration potential and regulating global climate change. The relationship between soil physical properties and soil organic carbon is not the main objective of this study. This study mainly considers the impact of soil physical properties on hyperspectral imagery and the bias it causes in SOM prediction results. Thank you very much for your valuable advice, for the relationship between soil physical properties and soil sustainability and carbon we will be more in-depth exploration in the future.

In addition to our point-by-point replies to the reviewers’ comments above, we also checked and modified the language of the full text. We did our best to meet the standards of required editorial corrections and have made all changes easily identifiable. We hope that our revised manuscript meets your requirements. If any further action is needed, please let us know immediately. We look forward to hearing back from you.

Thank you and best regards.

Yours sincerely,

The Authors

Round 2

Reviewer 1 Report

Comments and Suggestions for Authors

The authors have tried to incorporate most concerns from my previous review and I think it can be published.

Reviewer 2 Report

Comments and Suggestions for Authors

The authors have addressed all the comments adequately